# Electrochemical Impedance Spectroscopy on 2D Nanomaterial MXene Modified Interfaces: Application as a Characterization and Transducing Tool

**Juvissan Aguedo, Lenka Lorencova, Marek Barath** 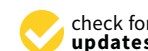**, Pavol Farkas and Jan Tkac \***

Institute of Chemistry, Slovak Academy of Sciences, 845 38 Bratislava, Slovakia; aguedo-ariza@savba.sk (J.A.); Lenka.Lorencova@savba.sk (L.L.); Marek.Barath@savba.sk (M.B.); Pavol.Farkas@savba.sk (P.F.)
\* Correspondence: Jan.Tkac@savba.sk; Tel.: +421-2-5941-0263

**Abstract:** This review presents the basic characteristics of MXene, a novel 2D nanomaterial with many outstanding properties applicable to electrochemical sensing and biosensing. The second part deals with electrochemical impedance spectroscopy (EIS) and its beneficial features applicable to ultrasensitive electrochemical sensing and label-free biosensing. The main part of the review presents recent advances in the integration of MXene to design electrochemical interfaces. EIS was used to evaluate the effect of anodic potential on MXene and the effect of the MXene preparation route and for characterization of MXene grafted with polymers. It also included the application of EIS as the main transducing tool for antibody- and aptamer-based biosensors or biosensors integrating molecularly imprinted polymers.

**Keywords:** MXene; electrochemical impedance spectroscopy (EIS); biosensors; nanomaterials; nanotechnology

---

## 1. 2D Nanomaterials—MXenes

MXenes were first described in 2011 [1] and proved to be the two-dimensional (2D) nanomaterials of the future. MXenes have been used in many applications due to their exceptional physical, chemical, and structural properties, which include good electrical conductivity, high surface area, hydrophilicity, biocompatibility, biodegradability, and ease of surface functionalization [2].

MXenes consist of carbides and nitrides of transition metals, with the formula of $M_{n+1}X_nT_x$, where "M" represents an early transition metal, "X" represents carbon or nitrogen, and "T" represents the terminal group (−F, −OH, and =O). Typically, MXenes are produced by selective chemical acid etching of the "A" layer from MAX phases producing MX phases. MAX phases have a general synthesized formula of $M_{n+1}AX_n$ ("A" represents an element from groups 13 and 14 of the periodic table) [3,4]. The MAX phases consist of repeating layers with a combination of a strong M-X bond and a weaker M-A bond. The "M" layer is arranged with the "A" layer, forming an octahedral structure with the X atoms in the interstice [5].

$Ti_3C_2$ MXenes were successfully synthesized by etching $Ti_3AlC_2$ with hydrofluoric acid (HF). This synthesis route follows three steps, the first being etching, followed by delamination and, finally, by intercalation. During the etching, the MAX phases are immersed in an HF solution; next, the delamination of the nanomaterial is caused by breaking the van der Waals forces between the MXene nanosheets. Finally, during intercalation, ions and/or molecules are inserted between the layers of laminated material to increase the interlayer distance. Nearly all types of MXenes with different

multi- or single-layered nanosheets can be fabricated by employing liquid-phase exfoliation, as shown in Figure 1a.

Since MXenes can be used in many applications, there is an ongoing need for improved methods for their synthesis. Alhabeb et al. [6] demonstrated that the LiF/HCl method made it possible to produce large MXene flakes of high quality. Moreover, a simple treatment of the nanomaterial with L-ascorbic acid can produce MXenes with increased electrical conductivity and enhanced stability against oxidation [7]. Fabrication of $Ti_3C_2T_x$-based devices remains challenging since interfacial $TiO_2$ groups could be oxidized, a process that is rapid in a liquid phase and slow in a solid state [8]. Furthermore, a combination of MXene with polymeric composites [9], nanohybrids [10], and nanoparticles [11] was investigated to enhance the stability of MXene [12] by preventing oxidation and enhancing shelf life. In an effort to improve the stability of this nanomaterial, for example, a spontaneous grafting of carboxy-betaine and sulpho-betaine on MXene was demonstrated as another solution [13]. Alternatively, storage of MXene in the form of a solution can be prolonged in hermetic argon-filled vials [14].

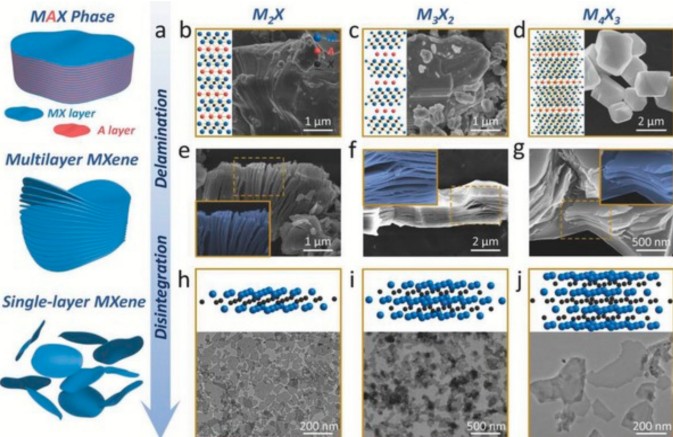

**Figure 1.** Methods for synthesis of MXenes for biomedical applications. (**a**) Synthesis of MXenes by delamination, disintegration, and, finally, by surface modification. (**b**) A 2D MAX phase model with corresponding scanning electron microscopy (SEM) image for $M_2X$, (**c**) 2D model and corresponding SEM image of MAX phase for $M_3X_2$, (**d**) 2D model and corresponding SEM image for $M_4X_3$ MAX phase, (**e**) SEM image of $M_2X$, (**f**) SEM image of $M_3X_2$, (**g**) SEM image of $M_4X_3$, and (**h**) 3D model with corresponding transmission electron microscopy (TEM) image for $M_2X$-based MXenes, (**i**) 3D model and corresponding TEM image for $M_3X_2$-based MXenes, and (**j**) 3D model and corresponding TEM image for $M_4X_3$-based MXenes. Reprinted from ref. [15], Copyright (2020), with permission from Elsevier.

Current studies on the biocompatibility and toxicity [15] of MXenes have shown enhanced biocompatibility after being modified with silk fibroin (SF) to obtain the SF@MXene bio-nanocomposite film working as a flexible pressure sensor (Figure 2) [16]. In addition, MXene was combined with MXene quantum dots [17] and collagen [15] in order to increase its stability.

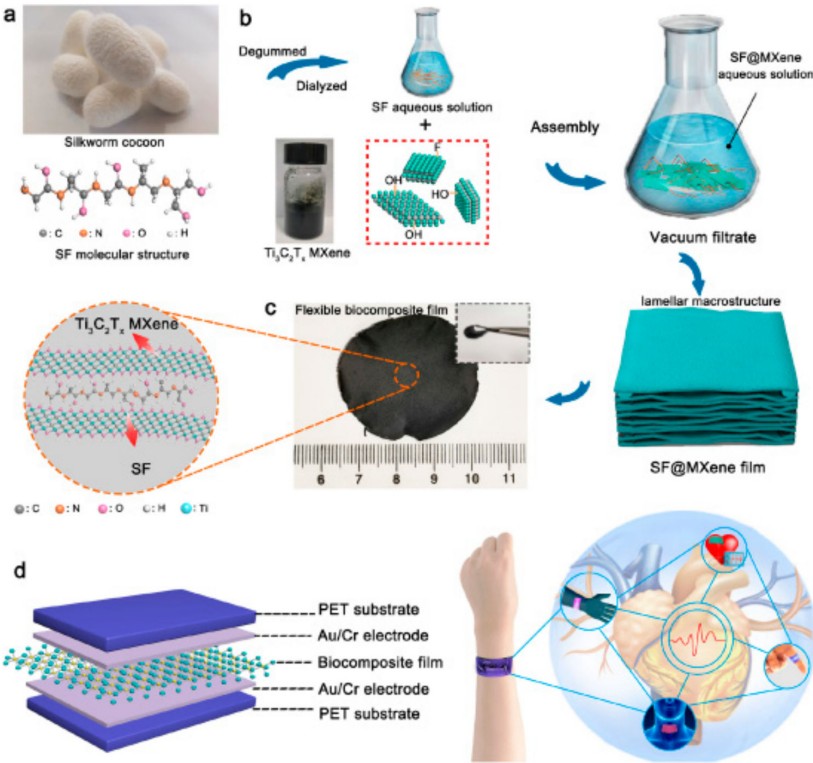

**Figure 2.** Preparation of the MXene-based bionanocomposite as a flexible pressure sensor. (**a**) Image of a silkworm cocoon with structure of a silk fibroin (SF) on a molecular level. (**b**) The fabrication of SF@MXene bionanocomposite film. (**c**) Image of SF@MXene bionanocomposite film. Inset shows SF@MXene bionanocomposite film on a molecular level. (**d**) Scheme of SF@MXene bionanocomposite film-based sensor applicable for monitoring of human health (right). Reprinted from ref. [16], Copyright (2020), with permission from Elsevier.

The physiological stability and biocompatibility of 2D $Nb_2C$ nanosheets were achieved in combination with polyvinylpyrrolidone without any noticeable toxicity exhibited either in vitro or in vivo. It was also noted that 2D MXenes are the candidates of choice as theranostic agents for cancer diagnostics and therapeutics [18]. Another type of MXene, i.e., $Ti_3C_2$ was also confirmed for in vitro/in vivo photothermal ablation of a tumor using near-infrared irradiation [19]. In addition, the authors demonstrated that biocompatible soybean phospholipid (SP)-modified $Ta_4C_3$ nanosheets [20], without any apparent toxicity, were effectively employed for in vitro/in vivo photothermal ablation of a tumor. $Ti_3C_2T_x$ MXene was proven as a biocompatible and safe nanomaterial during examination of its toxicity using zebrafish embryos as an in vivo model [21]. A multifunctional composite consisting of $Ti_3C_2$ patterned with paramagnetic $MnO_x$ [22] species can play an important role in theranostic medicine, e.g., for efficient magnetic resonance and photoacoustic dual-modality imaging-guided photothermal therapy to cure cancer. $Nb_2C$ quantum dots (QDs) [23] behave in a more biocompatible way to human umbilical vein endothelial cells (HUVECs) than $Ti_3C_2$ QDs, which show a cytotoxic reaction. The 2D $Ti_2NT_x$ nanosheets [24] exhibited higher toxicity toward cancerous cell lines than normal ones with no toxic behavior to nonmalignant cell lines.

To date, 20+ different MXene compositions have been synthesized, comprised of $Ti_3C_2T_x$, $Ti_2CT_x$, $Ti_2NT$, $Mo_2CT_x$, and/or $Nb_4C_3T_x$ [25], 70 varieties of MAX phases have been investigated, and more MAX phases are predicted [2]. In the last 10 years, MXenes have become a very attractive material for use in a variety of applications, including biosensors [26,27], sensors with improved electrocatalytic activity [4,28–30], wearable sensors [31–34], MXene-based flexible microelectrode arrays for recording of neural activity [35,36], biomedical applications [37,38], energy storage technologies [39], supercapacitors [40], batteries [41], hydrogen storage devices [42],

catalytic devices [43], photocatalysts [44], field-effect transistors [45], water purification systems [46], devices for environmental remediation [47], and, in addition, due to their outstanding conductivity [48], they have been employed as an electromagnetic shielding material [49].

## 2. Electrochemical Impedance Spectroscopy (EIS): The Right Tool for Investigating Conductive Interfaces

Electrochemistry as an interdisciplinary research area investigates electrical phenomena by chemical reactions on conductive interfaces. Since the 1990s, nanomaterials (nanoconcrete, nanofilms, nanoparticles, nanoshells, nanospheres, nanotubes, nanowires, nanoclays, and nanopolymers), due to their unique multifunctional (catalytic, electronic, chemical, magnetic, mechanical, optical, physical) properties, have become key components in numerous applications.

Electrochemical methods (cyclic voltammetry (CV), chronoamperometry, chronopotentiometry, polarographic techniques, pulse methods, electrochemical impedance spectroscopy (EIS), etc.) can be effectively applied to characterization of the interfacial properties of nanomaterials [50]. This is made possible since electrochemical methods are quick, reliable, and sensitive to interfacial changes on a nanoscale level. In particular, electrochemistry can provide information on the redox stability of nanomaterials, stability against oxidation, and resistance to biofouling, and it is also possible to monitor the success of surface modification by the use of nanomaterials. At the same time, it should be noted that such changes can be monitored in situ and in real time if required.

The electron-transfer properties of the modified conductive surfaces and associated interfacial chemical transformations can be evaluated using EIS. This technique provides mechanistic and kinetic information about conductive surfaces as key components of batteries and fuel cells and for the characterization of various coatings, corrosion processes, semiconducting electrodes, and solid-state systems.

EIS provides valuable information on whatever system is under examination, such as the resistance of the electrolyte between a working electrode (WE) and a reference one, interfacial charge transfer resistance on WE due to adsorbed species (ions, organic, and water molecules), charge transfer resistance due to Faradaic processes, diffusion processes (Warburg factor), and adsorption of the reactant/product on the interface. Such complicated redox behavior can be described by an equivalent circuit containing resistance, inductance, and capacitance contributions (Figure 3) [50].

An alternating current (AC) is employed to perturb redox processes on a WE. The EIS setup consists of the electrochemical cell, an impedance analyzer, and a computer serving to collect data. In order to obtain the EIS spectra, the conductive interface is exposed to sinusoidal AC voltage excitations with low amplitude and the resultant response of the system is recorded as the amplitude and phase of the current. The frequency of the AC signal applied to the system is varied with the overall impedance of the cell recorded as a function of an input frequency, while the impedance is monitored. Hence, it is possible to identify processes occurring on different timescales and to separate electronic from ionic processes in mixed conductors.

The impedance is represented as a complex quantity $Z$, i.e., $Z(\omega) = Z_{Re} - jZ_{Im}$ consisting of a real part ($Z_{Re} = R$, where $R$ is resistance) and an imaginary part ($Z_{Im} = 1/\omega C$, where $C$ is capacitance and $\omega$ is the angular frequency). The impedance facilitates the description of complex circuits by graphical representation by using two different plots: The first describes imaginary *vs.* real impedance at different frequencies and the second describes absolute impedance *vs.* frequency, titled the Nyquist plot (complex plane plot) and the Bode plot, respectively. In comparison to the Nyquist plot, the Bode plot interprets impedance more clearly as a function of a frequency [51].

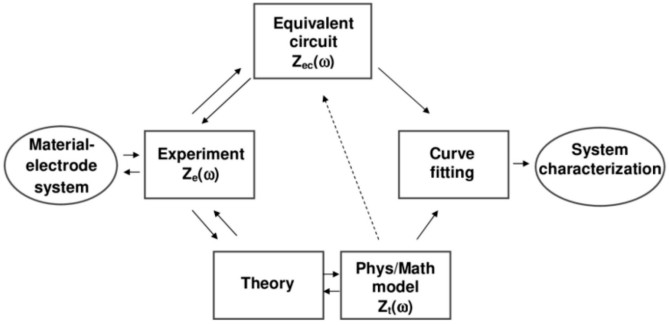

**Figure 3.** A scheme showing the use of electrochemical impedance spectroscopy (EIS) to investigate characteristics of a material–electrode interface. Reprinted from ref. [52], Copyright 2005, with permission from John Wiley and Sons.

A Nyquist plot, having the real impedance component ($Z'$) on the *x*-axis, and the imaginary component ($Z'$) on the *y*-axis, is fitted with equivalent electrical circuits to obtain the device properties in a quantitative way, as depicted in Figure 4 [53]. Moreover, the shape of the Nyquist plot provides information on the physical processes occurring on the conductive interfaces. For example, a line parallel to the *y*-axis in the Nyquist plot means that the equivalent circuit consists of only a capacitor *C* (Figure 4b) or a capacitor *C* in series with a solution resistor $R_s$ (Figure 4c). If the Nyquist plot has a semicircular shape with an intercept, the equivalent circuit consists of a capacitor *C* in parallel with the interfacial resistor *R* (Figure 4d) or a capacitor *C* in parallel with the solution resistor *R* in series with solution resistance $R_s$ (Figure 4e). Visual examination of the Nyquist plot combined with knowledge of the device makes it possible for an equivalent circuit model to be set up accurately, correctly detailing all the processes involved (Figure 4f,g). Data present in the Nyquist plot close to the origin are associated with the high-frequency spectrum.

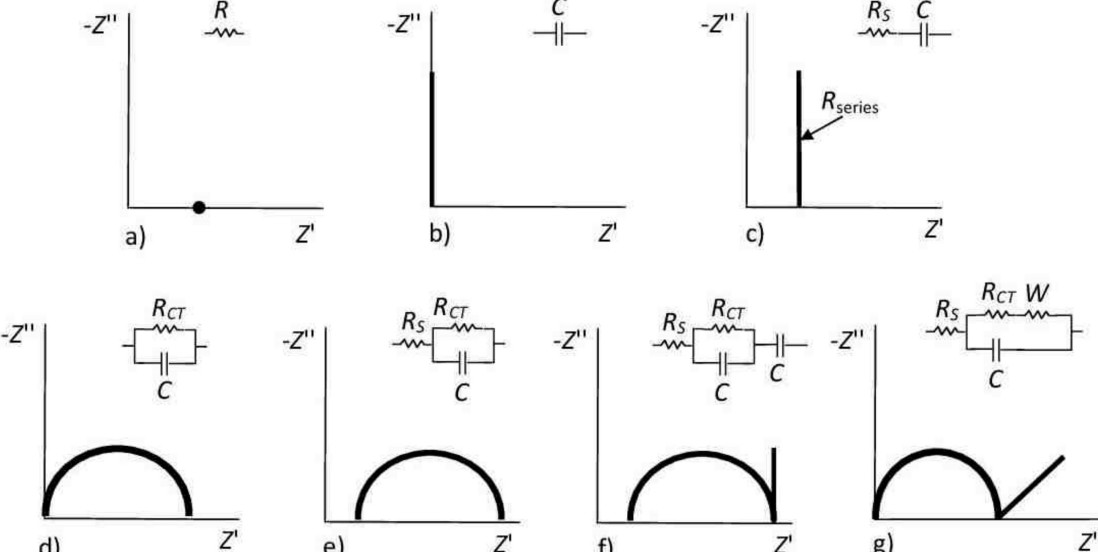

**Figure 4.** Typical Nyquist plots for various electrical circuits, moving from those that are the most simple to the more common. Upper row: (**a**) pure resistor (*R*, only real part of impedance); (**b**) pure capacitor (*C*, only imaginary part of impedance); (**c**) *R* and *C* in series connection. Lower row: (**d**) *R* and *C* in parallel connection; (**e**) *R* and *C* in parallel connection and in series with a solution resistance ($R_S$), where $R_{CT}$ denotes the charge transfer resistance of the interface; (**f**) the same circuit with additional in-series capacitance and (**g**) Randles–Ershler equivalent circuit, most commonly used for fitting impedimetric biosensor data, where *W* stands for Warburg element (diffusion parameter for low frequencies). Adapted from ref. [54], Copyright (2014), with permission from Elsevier and reprinted from ref. [51], Copyright 2019, with permission from John Wiley and Sons.

On the other hand, if EIS is applied without understanding of the basic interfacial processes, it is difficult to interpret the data correctly and to choose a proper equivalent circuit.

Thus, circuit elements identified upon the fitting of a Nyquist plot can provide both qualitative and quantitative data, such as a time constant or an activation energy for each particular process [53]. Analysis of the EIS data is not always straightforward. For instance, redox properties on rough surfaces cannot be properly described with a capacitive element only, which is why a constant phase element (CPE) is used instead. For an extensive overview of the fundamental and applied aspects of EIS, the reader is referred to the literature [52].

## 3. Electrochemical Investigation of MXene-Modified Interfaces via EIS

EIS can be beneficially applied to the characterization of MXene interfaces using various redox probes or working electrolytes [55–58]. In order to prepare effective electrochemical (bio)sensors based on MXene-modified electrodes, it is most important to understand the electrochemical behavior of such a nanomaterial-modified electrode in an anodic and a cathodic potential window. In our initial study, the electrochemical stability window for $Ti_3C_2Tx$ MXene-modified glassy carbon electrode surface ($GCE/Ti_3C_2Tx$) was examined [59]. In a subsequent study we investigated the electrochemical behavior of electrodes patterned by MXene prepared from the MAX phase using two distinct methods (HF or LiF/HCl etching) [29]. Our third study described the application of EIS to the characterization of MXene-modified interfaces grafted with polymers, which resisted nonspecific protein binding [13].

### 3.1. EIS for Characterization of Redox Stability of MXene-Modified Electrodes and Sensors

The $Ti_3C_2T_x$MXene nanomaterial is prone to anodic oxidation when exposed to a working potential above +200 mV *vs.* Ag/AgCl reference electrode [59]. In order to better characterize this process, a number of techniques including electrochemical ones were used to study the electrochemical oxidation of MXene. One of the electrochemical methods used was EIS. For this examination, a $R_s(Q[R_{ct}W])$ Randles equivalent circuit consisting of an electrolyte resistance $R_s$, constant phase element $Q$, charge transfer resistance $R_{ct}$, Warburg element $W$, and 5 mM ferri/ferrocyanide redox couple probe was used. The charge transfer resistance ($R_{ct}$) values obtained for unmodified GCE, pristine $Ti_3C_2T_x$ (T: =O, −OH, −F)-modified GCE, and oxidized $Ti_3C_2T_x$-modified GCE were (164 ± 36) Ω, (7130 ± 600) Ω, and (8500 ± 1000) Ω, respectively (Figure 5 left). The results demonstrated increased resistivity on the part of the less electrochemically active $oTi_3C_2T_x$ (i.e., electrochemically oxidized $Ti_3C_2T_x$) compared to $Ti_3C_2T_x$, indicating that $oTi_3C_2T_x$ is less conductive and less electrochemically accessible than $Ti_3C_2T_x$. The study also indicates that exposure of the MXene-modified GCE electrode to a working potential exceeding +200 mV *vs.* Ag/AgCl induces the formation of $TiO_2$ particles at an interface between a solution and an outer MXene layer with the subsequent dissolution of such particles by $F^-$ ions. This process is irreversible from an electrochemical perspective. On the other hand, the MXene-modified interface is stable in the cathodic potential window with the application of such an interface for the reduction of $H_2O_2$ or $O_2$. Finally, the MXene-modified electrode was employed for the ultrasensitive detection of $H_2O_2$ in the cathodic potential window with a limit of detection (LOD) of 0.7 nM and with a sensitivity of 596 mA cm$^{-2}$ mM$^{-2}$ [59].

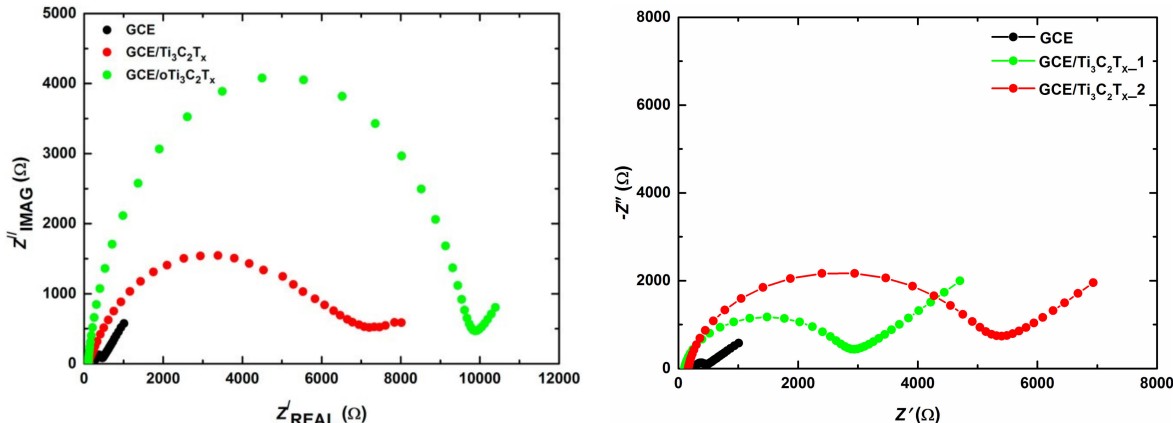

**Figure 5.** Nyquist plots at unmodified glassy carbon electrode (GCE), GCE/Ti$_3$C$_2$T$_x$, and GCE/oTi$_3$C$_2$T$_x$ (**left**). Reprinted from ref. [59], Copyright (2017), with permission from Elsevier. Nyquist plots at unmodified GCE, GCE/MXene1, and GCE/MXene2 (**right**). Reprinted by permission from ref. [29]: Springer, Copyright (2020). In both cases EIS was run in 5 mM ferricyanide/ferrocyanide solution in 0.1 M PB pH 7.0. EIS spectra were run using 50 different frequencies (0.1 Hz–100 kHz). The experiment was performed using an Ag/AgCl reference electrode and platinum counter electrode.

Huang et al. [60] demonstrated the use of an electrochemical sensor employing titanium carbide (Ti$_3$C$_2$) and a multi-walled, carbon nanotubes (MWCNTs)-based nanocomposite for the detection of catechol and hydroquinone. The nanocomposite was prepared by blending Ti$_3$C$_2$ MXene and MWCNTs *via* a self-assembled 2D hierarchical process. The analytes were detected with LOD of 6.6 nM for hydroquinone and of 3.9 nM for catechol (a linear range of 2–150 μM for both analytes). Recovery values of 96.9%–104.7% and 93.1%–109.9% were obtained for hydroquinone and catechol detection in industrial wastewater samples. EIS was implemented for the analysis of $R_{ct}$ values on all the modified electrodes. The $R_{ct}$ value of 131 Ω for the bare GCE electrode was the largest. The $R_{ct}$ values of MXene/GCE and MWCNTs/GCE were smaller, i.e., 119 Ω and 105 Ω, respectively. The decrease in the $R_{ct}$ value confirmed that the presence of MXene or MWCNTs improved the electrical conductivity of the modified electrode. After the introduction of both nanomaterials, the $R_{ct}$ value decreased even further to 97 Ω at MXene-MWCNTs/GCE [60].

It may be concluded that EIS is an effective tool for the in situ characterization of the charge transfer resistance (or conductivity) of MXene-modified interfaces applied as sensors. In particular, EIS can provide information as to whether MXene was exposed to the potential leading to MXene oxidation, as well as information on whether it is worth combining MXene with other types of nanomaterials in order to render such a hybrid nanoparticle-modified interface more conductive.

### 3.2. EIS for Redox Characterization of Electrodes Modified with MXenes Prepared Using Two Etching Procedures

There are several applicable routes for the preparation of MXene from the MAX phase, either using HF, in situ-generated HF (from a mixture of LiF and HCl), or other methods [5]. Although the basic physico-chemical parameters of MXene prepared *via* two distinct routes (i.e., HF or LiF/HCl etching) have been extensively elucidated in a number of studies (see, for example, [5] and the references cited therein), the electrochemical behavior of MXenes prepared in two different ways has only recently been examined by our group [29].

Significant differences in the redox interfacial behavior of Ti$_3$C$_2$T$_x$ MXenes prepared by using two etchants, i.e., HF (MXene1) or LiF/HCl (MXene2), were observed [29]. Notably, MXene1-modified GCE did not exhibit a large anodic oxidation peak, while MXene2-modified GCE showed a significantly larger anodic peak. The study by applying redox probe (Ru(NH$_3$)$_6$Cl$_3$ exhibited a higher negative charge density present in MXene2 than in MXene1. This suggests that, from an electrochemical perspective, MXene1 is oxidized with a low density of a negative charge. The application of EIS for

interfacial characterization demonstrated that $R_{ct}$ was higher for GCE/MXene2 (5.0 ± 0.2) kΩ than for GCE/MXene1 (2.6 ± 0.3) kΩ. The inner sphere redox probe (ferricyanide/ferrocyanide redox couple) indicated a higher density of negatively charged species present on the surface of GCE/MXene2 than on GCE/MXene1 (Figure 5 right) and this may be why the MXene2-modified GCE exhibits higher $R_{ct}$ than the MXene1-modified GCE. The study revealed a remarkable difference in the redox properties of the MXene-modified interfaces, which are significantly influenced by the way MXene was prepared from the MAX phase. Finally, the MXene2-modified GCE was significantly more electrochemically active toward the reduction of $H_2O_2$ (215-fold) and $O_2$ (317-fold) than the MXene1-modified GCE [29].

The EIS investigation was used successfully to better understand the interfacial properties of MXenes prepared *via* two distinct routes. In particular, EIS not only provided insights into the initial redox conditions but also about the density of the negative charge on MXenes prepared using two routes.

### 3.3. EIS for Electrochemical Characterization of MXene-Modified Electrodes with Grafted Polymers

In order to render the MXene-modified interfaces applicable to biosensing and make them available to other advanced applications, interfacial patterning of the MXene should be performed [13]. In the previous sections, it was shown that MXene prepared *via* the LiF/HCl method contains a significant negative charge, resembling the presence of plasmons (free electron cloud) in metallic nanoparticles such as gold nanoparticles [61,62]. Such free electrons can be used for the spontaneous reduction of polymers containing a diazonium functional group. If such a polymer also contains betaine functional groups in addition to the diazonium functional group, then the polymer grafted onto MXene can resist nonspecific protein binding [13].

In order to achieve this aim, the modification of $Ti_3C_2T_x$ MXene using aryldiazonium-based grafting with derivatives bearing a sulpho-(SB) or carboxy-(CB) betaine pendant moiety was performed. MXene contains free electrons; hence, betaine derivatives can be grafted onto MXene spontaneously. Spontaneous SB and CB grafting kinetics toward MXene were investigated electrochemically confirming more rapid SB grafting than CB grafting. A number of electrochemical methods used to examine non-Faradaic and Faradaic interfacial electrochemical properties using two redox probes (ferri/ferro redox couple and $Ru(NH_3)_6^{3+}$ redox probe) as well as contact-angle measurements and secondary ion mass spectrometry (SIMS) analysis revealed significant differences for the betaine layers spontaneously grafted onto MXene. In addition to the spontaneous CB and SB grafting onto MXene, grafting induced by a redox trigger was also tested. A denser interfacial layer of CB and SB on the MXene was obtained *via* electrochemical grafting than by the spontaneous process. Moreover, an electrochemically grafted SB layer exhibited a much lower $R_{ct}$ value and an electroactive surface area than the CB layer prepared *via* an electrochemical grafting. Hence, the redox behavior of an MXene-modified interface can effectively be fine-tuned by adjusting the SB/CB ratio in the solution during electrochemical grafting. EIS analysis revealed that CB-modified MXene exhibited a higher $R_{ct}$ value (i.e., 838 Ω for MXene/CB/GCE and 1610 Ω for MXene/CB_e/GCE) than for pristine MXene/GCE (786 Ω) (Figure 6). Conversely, SB-modified MXene interfaces exhibited much lower $R_{ct}$ values (i.e., 88 Ω for MXene/SB/GCE and 228 Ω for MXene/SB_e/GCE) than for unmodified MXene/GCE (786 Ω) (Figure 6).

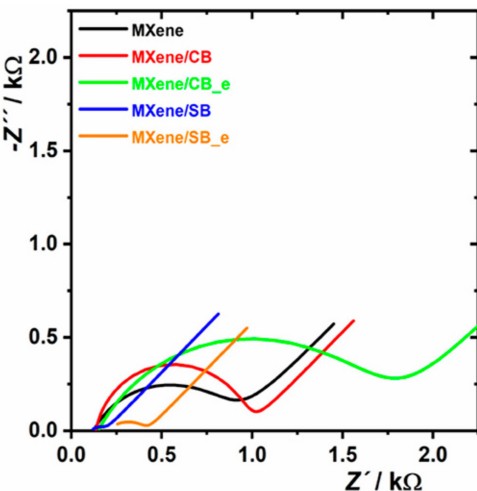

**Figure 6.** EIS investigation performed on MXene-modified electrode and on the electrode patterned by zwitterion-modified MXene. For other experimental conditions see Figure 5. EIS analysis is shown as a Nyquist plot using an equivalent circuit $R[Q(RW)]$ for data fitting. Abbreviations: MXene/CB: MXene spontaneously modified by carboxybetaine; MXene/SB: MXene spontaneously modified by sulphobetaine; MXene/CB_e: MXene modified by carboxybetaine using a redox trigger; MXene/SB_e: MXene modified by sulphobetaine using a redox trigger. The figure is reprinted from our open access paper [13].

EIS investigation indicated an effective pairing of the anionic and cationic groups within the SB layer resulting in low resistance toward a ferri/ferro redox couple. The study also revealed that zwitterion-modified MXene prepared by electrochemical grafting exhibited higher $R_{ct}$ values than those recorded in the spontaneous process for CB (1610 Ω *vs.* 838 Ω) and for SB (228 Ω *vs.* 88 Ω), as well. It was proved that a higher density of zwitterions on the interface was induced electrochemically than in the process based on spontaneous grafting [13].

## 4. Electrochemical Investigation of MXene-Modified Bio-Interfaces via EIS

EIS is an ideal tool for characterizing the step-by-step build-up of a biosensing interface consisting of several subsequent incubation steps since such a modification might be monitored directly on the conductive interface in situ after each step. Accordingly, EIS is an important tool for characterizing the incremental steps needed for the successful development of biosensors.

### 4.1. EIS for Electrochemical Investigation of MXene-Modified Enzyme Biosensors

Zhou et al. [63] introduced a novel biosensor based on MXene and the acetylcholinesterase (AChE) enzyme for the detection of organophosphorus pesticides. In the first step, pretreated GCE was activated with 0.5 M $H_2SO_4$ then subsequently modified with the drop-casting method with dispersion consisting of $Ti_3C_2T_x$ MXene nanosheets and chitosan (CS) in order to obtain CS-MXene/GCE. The CS-MXene nanocomposite, due to its high surface area, biocompatibility, nontoxicity, and the ability to form films, was used as a support to immobilize the enzyme for preparation of the AChE biosensor with enhanced activity and stability. This outstanding performance was also accomplished due to the excellent electron conductivity of the CS-MXene nanocomposites.

The biosensor could detect malathion with LOD of $0.3 \times 10^{-14}$ M in a linear range of $1 \times 10^{-14}$ M–$1 \times 10^{-8}$ M [63]. The interfacial electrode properties during the stepwise preparation were examined by EIS. The CS-MXene/GCE exhibited much lower $R_{ct}$ than the bare GCE. It was confirmed that CS-MXene nanocomposites could significantly increase the electron transfer and enhance the diffusion of the redox probe toward the interface due to the high conductivity and high surface area attributed to the bio-nanocomposite. The $R_{ct}$ of the AChE/CS-MXene/GCE increased in comparison

with CS-MXene/GCE, confirming that the presence of an enzyme layer is an additional barrier between the electrode and electrolyte (Figure 7) [63].

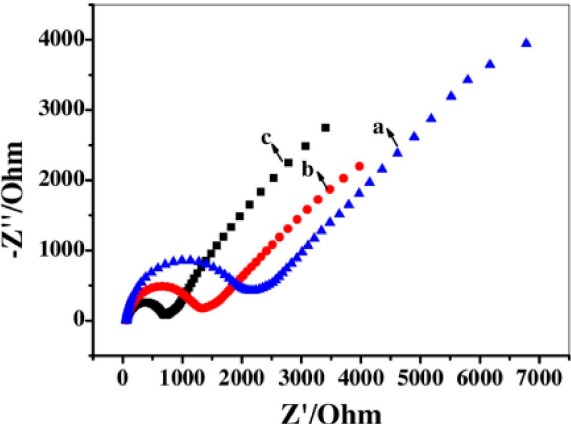

**Figure 7.** Nyquist plots recorded for unmodified GCE (a), AChE/CS-MXene/GCE (b), CS-MXene/GCE (c) in 5 mM $K_3[Fe(CN)_6]$ and 0.1 M KCl mixture solution over a frequency range from $10^{-2}$ Hz to $10^5$ Hz. CS stands for chitosan. Reprinted from ref. [63], Copyright (2017), with permission from Elsevier.

Two electrochemical methods (EIS and CV) were used to observe the build-up of the biosensing interface [63]. Both methods provided the same conclusion that modification of GCE with CS-MXene made the system more conductive while, after immobilization of the enzyme, the conductivity of the interface slightly dropped (Figure 7). This latter confirmed the successful immobilization of the enzyme on the interface.

### 4.2. EIS for Electrochemical Investigation of MXene-Modified Immunosensors

Medetalibeyoglu and coworkers developed an ultrasensitive electrochemical antibody-based biosensor for the detection of a prostate-specific antigen (PSA) with LOD of 3.0 fg mL$^{-1}$ and a linear range of 0.01–1.0 pg mL$^{-1}$ [64]. Firstly, the pretreated GCE electrode surface was modified with gold nanoparticles patterned by *p*-aminothiophenol (AuNPs-ATP) and further functionalized with graphene oxide (GO). In the next step, the hybrid nanocomposite was modified by adsorption of the primary anti-PSA antibodies (Ab$_1$). In order to eliminate nonspecific protein binding, the hybrid nanocomposite was blocked with bovine serum albumin (BSA). The biosensor device thus derived was then ready to be incubated with the analyte (PSA). Subsequently, an anti-PSA secondary antibody (Ab$_2$) immobilized on MXene/AuNP was incubated with the modified electrode, making a sandwich configuration. EIS was used as a useful characterization method [64]. The successful immobilization of proteins such as Ab$_1$, PSA antigen, and BSA in individual steps during the biosensor assembly was confirmed by the EIS technique. EIS analysis demonstrated a decrease in resistance values after modification of GCE with GO and the AuNPs-ATPGO composite. The subsequent attachment of Ab$_1$ to AuNPs-ATPGO/GCE resulted in an increase in resistance due to the low conductivity of Ab$_1$. The resistance of the interface increased after blocking with BSA and incubation with PSA. EIS confirmed that the construction of the PSA/BSA/Ab$_1$/AuNPs-ATPGO/GCE biosensor was successfully accomplished (Figure 8) [64].

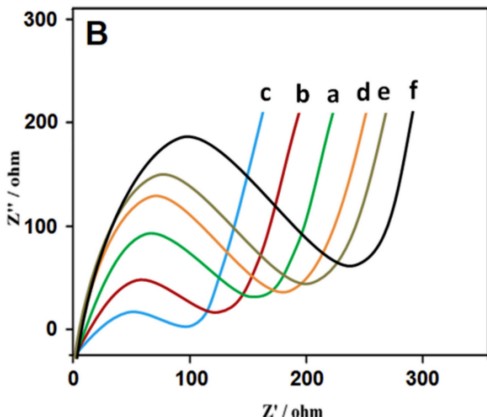

**Figure 8.** Nyquist plots at (**a**) unmodified GCE, (**b**) GO/GCE, (**c**) AuNPs-ATP-GO/GCE, (**d**) after immobilization of $Ab_1$ on AuNPs-ATP-GO/GCE, (**e**) after blocking of the interface by BSA, and (**f**) after incubation with PSA. Abbreviations: GO: graphene oxide; AuNPs: gold nanoparticles; ATP: *p*-aminothiophenol; BSA: bovine serum albumin; PSA: prostate specific antigen. For other abbreviations see the text. Reprinted from ref. [64], Copyright (2020), with permission from Elsevier.

Medetalibeyoglu et al. developed an ultrasensitive electrochemical sandwich-type immunoassay based on antibody-antigen recognition for procalcitonin (PCT) determination. PCT is an important biomarker related to septicemia diagnostics and for the detection of bacterial inflammations [65]. PCT (molecular mass of 13 kDa), composed of 116 amino acids, is a precursor of the hormone calcitonin, which is produced by cells in the thyroid and the neuroendocrine cells of the lung. Its amount increases, especially as a result of a pro-inflammatory stimulus. In the first step, delaminated sulphur-doped MXene (S-MXene) with =O, −OH, −F terminal groups was synthesized by etching the aluminium layer out of the polycrystalline sulphur-doped MAX phase powder ($S-Ti_3AlC_2$) in 6.0 M HCl/LiF (in situ formation of HF). This dispersion was deposited onto the pretreated GCE surface (d-S-MXene/GCE) and, after drying using infrared irradiation, the d-S-MXene/GCE was immersed into an aqueous dispersion of AuNPs, resulting in the preparation of the AuNPs/d-S-MXene/GCE composite *via* a sulphur-gold affinity between S-MXene and AuNPs. Preparation of the immunosensor was completed by immobilization of the monoclonal primary antibodies $Ab_1$, and the surface was blocked with BSA to resist nonspecific interactions. The bioaffinity between the PCT and $Ab_1$ was exploited for the determination of PCT. Finally, the electrode was incubated with the bio-nanocomposite containing a secondary antibody $Ab_2$ attached to carboxylated graphitic carbon nitride (c-g-$C_3N_4$). The signal for the detection of PCT was generated by c-g-$C_3N_4$ having good electrocatalytic activity toward $H_2O_2$. The fabricated immunosensor was able to detect PCT with LOD of 2.0 fg $mL^{-1}$ within a linear range of 0.01–1.0 pg $mL^{-1}$. Out of a number of microscopic and spectroscopic techniques available, EIS provided useful information about the successful development of a biosensor in a step-by-step fashion [65].

EIS (Figure 9B), as the tool of choice to examine the interfacial properties of modified, conductive layers was also independently confirmed by CV (Figure 9A,C) [65]. As shown in the Nyquist diagram, the $R_{ct}$ decreased after modification of GCE with delaminated sulphur-doped MXene (i.e., d-S-MXene/GCE) and AuNPs/d-S-MXene/GCE in comparison with bare GCE. The $R_{ct}$ increased after immobilization of $Ab_1$ on AuNPs/d-S-$Ti_3C_2T_x$MXene/GCE. After blocking by BSA, the $R_{ct}$ increased and then further increased after incubation with the analyte (PCT) [65].

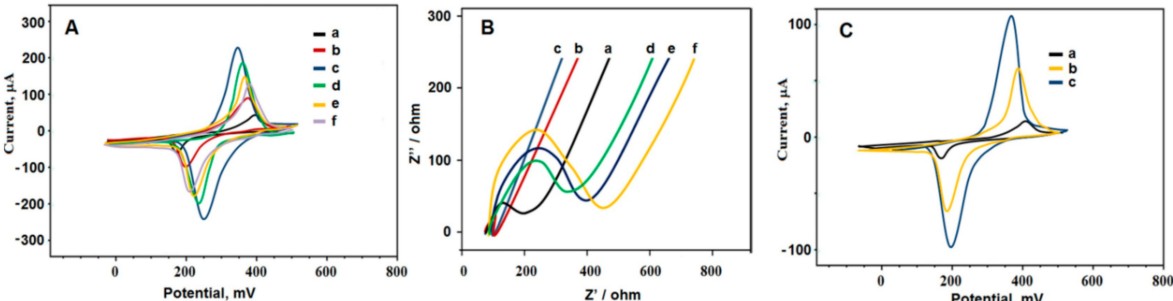

**Figure 9.** (**A**) Cyclic voltammograms (CVs) at (**a**) unmodified GCE, (**b**) d-S-MXene/GCE, (**c**) AuNPs/d-S-MXene/GCE, (**d**) Ab$_1$/AuNPs/d-S-MXene/GCE, (**e**) BSA/Ab$_1$/AuNPs/d-S-MXene/GCE, (**f**) PCT/BSA/Ab$_1$/AuNPs/d-S-MXene/GCE; (**B**) EIS response at (**a**) unmodified GCE, (**b**) d-S-MXene/GCE, (**c**) AuNPs/d-S-MXene/GCE, (**d**) Ab$_1$/AuNPs/d-S-MXene/GCE, (**e**) BSA/Ab$_1$/AuNPs/d-S-MXene/GCE, (**f**) PCT/BSA/Ab$_1$/AuNPs/d-S-MXene/GCE; (**C**) CVs at (**a**) unmodified GCE, (**b**) d-MXene/GCE, (**c**) d-S-MXene/GCE. BSA stands for bovine serum albumin and PCT stands for procalcitocin. Reprinted from ref. [65], Copyright (2020), with permission from Elsevier.

### 4.3. EIS for Electrochemical Investigation of MXene-Modified Aptasensors

Zhou et al. [66] developed an impedimetric aptamer biosensor based on a nanostructured multicomponent hybrid consisting of Ti$_3$C$_2$T$_x$ MXene nanosheets and phosphomolybdic acid nanoparticles (PMo$_{12}$) integrated within a polypyrrole (PPy) matrix (PPy@Ti$_3$C$_2$T$_x$/PMo$_{12}$) for the detection of osteopontin (OPN). The aptamers were immobilized *via* the $\pi - \pi$ stacking, electrostatic interactions, and hydrogen bonds on such a nanocomposite. The $R_{ct}$ values obtained were 50 $\Omega$, 0.35 k$\Omega$, 1.55 k$\Omega$, and 2.35 k$\Omega$ for the bare gold electrode (AE), the PPy@Ti$_3$C$_2$T$_x$/PMo$_{12}$/AE, the Apt/PPy@Ti$_3$C$_2$T$_x$/PMo$_{12}$/AE, and the OPN/Apt/PPy@Ti$_3$C$_2$T$_x$/PMo$_{12}$/AE electrodes, respectively. The increase in $R_{ct}$ value confirmed modification of the bare electrode with the nanostructured hybrid, immobilization of the aptamer, and detection of OPN (Figure 9). In addition, EIS can be used effectively as a transducing platform for the robust detection of cancer biomarkers. The change in $R_{ct}$ values was used to compare the efficiency of the aptamer immobilization and to evaluate the sensitivity of the OPN detection using various aptasensors (Figure 10g). As demonstrated in the Nyquist plots, the simulated $R_{ct}$ values gradually increased with the increasing OPN concentration (Figure 11). The PPy@Ti$_3$C$_2$T$_x$/PMo$_{12}$-based aptasensor exhibited LOD of 0.98 fg mL$^{-1}$ (i.e., ~15 aM; signal to noise ratio; *S/N* 3, a linear range of 0.05–10,000 pg mL$^{-1}$) by measuring the dependence of $\Delta R_{ct}$ on different concentrations of OPN. In addition, the biosensor exhibited good selectivity for analyte detection, reproducibility of the assay, and storage stability and reusability of the biosensor using surface regeneration (Figure 11c,f) [66].

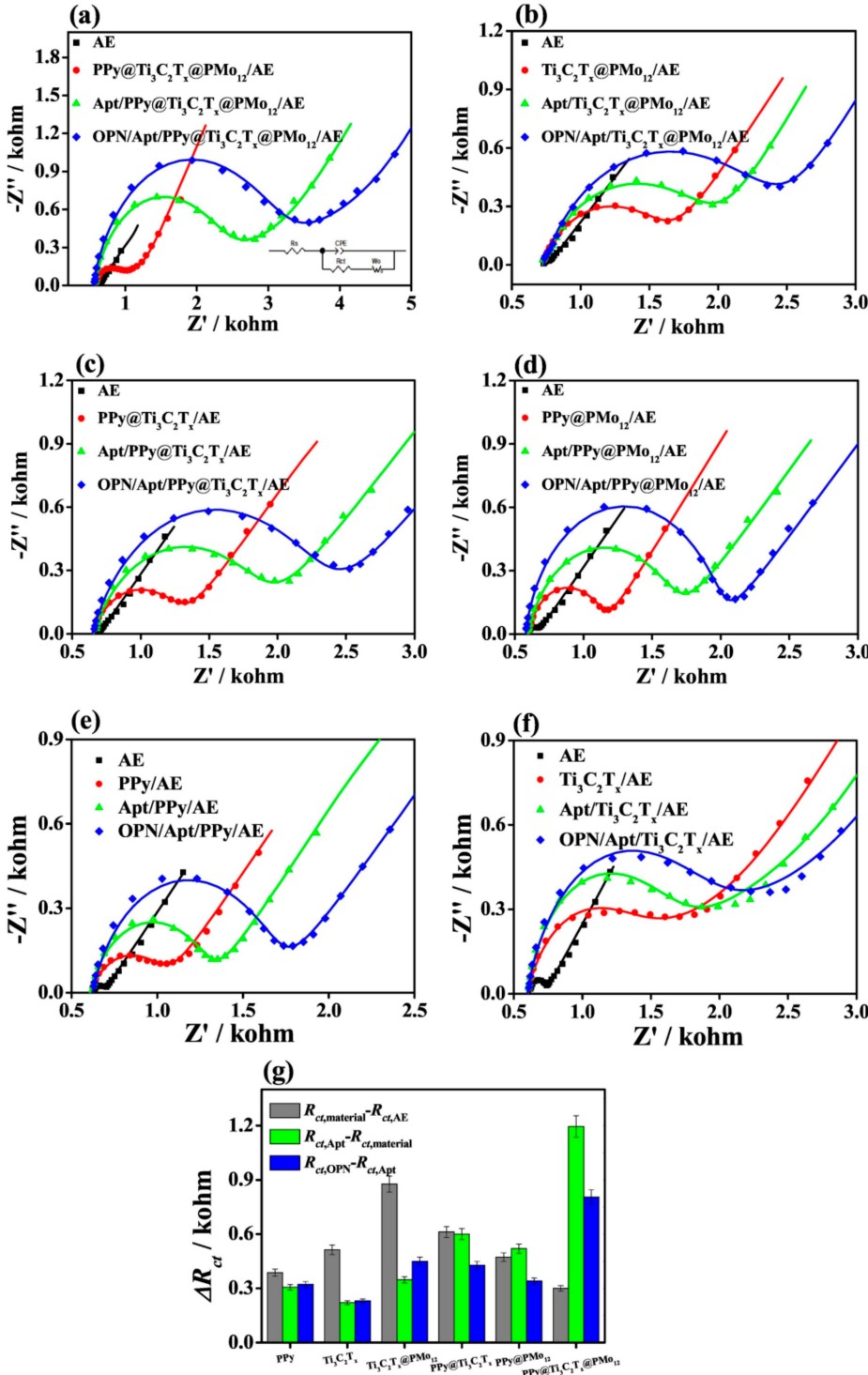

**Figure 10.** Nyquist plots for (**a**) PPy@Ti$_3$C$_2$T$_x$/PMo$_{12}$-, (**b**) Ti$_3$C$_2$T$_x$@PMo$_{12}$-, (**c**) PPy@Ti$_3$C$_2$T$_x$-, (**d**) PPy@PMo$_{12}$-, (**e**) PPy-, and (**f**) Ti$_3$C$_2$T$_x$-based aptasensors and (**g**) their corresponding variations in $\Delta R_{ct}$ values at each stage for the OPN detection using the fabricated aptasensors based on PPy, Ti$_3$C$_2$T$_x$, Ti$_3$C$_2$T$_x$@PMO$_{12}$, PPy@Ti$_3$C$_2$T$_x$, PPy@PMo$_{12}$, and PPy@Ti$_3$C$_2$T$_x$/PMo$_{12}$ nanohybrids. PPy stands for polypyrrole; PMo$_{12}$ stands for phosphomolybdic acid nanoparticles and OPN stands for osteopontin. Reprinted from ref. [66], Copyright (2019), with permission from Elsevier.

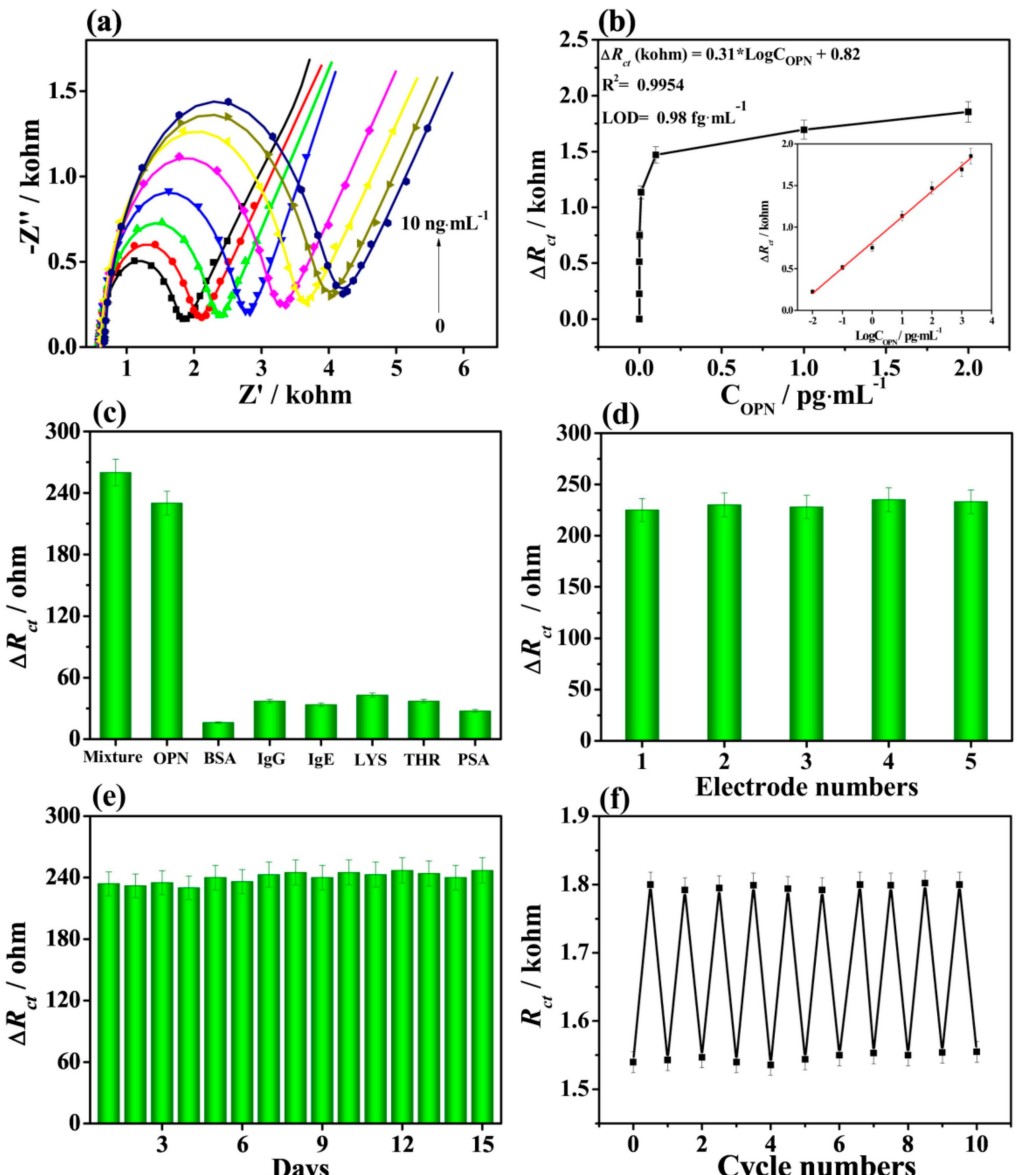

**Figure 11.** (**a**) Nyquist plots for the biosensor detecting OPN with concentrations from 0.05 to 10,000 pg mL$^{-1}$. (**b**) Dependence of the $\Delta R_{ct}$ values on OPN concentration. (**c**) Interference study using BSA, immunoglobulin G (IgG), immunoglobulin E (IgE), lysine (LYS), thrombin (THR), and prostate specific antigen (PSA) as interferents applied at concentration of 50.0 pg mL$^{-1}$, OPN at concentration of 0.5 pg mL$^{-1}$, and their mixture. (**d**) Reproducibility, (**e**) stability, and (**f**) regeneration of the biosensor for detection of OPN (0.5 pg mL$^{-1}$). Reprinted from ref. [66], Copyright (2019), with permission from Elsevier.

### 4.4. EIS for Electrochemical Investigation of MXene-Modified Biosensors for Detection of micro RNA (miRNA)

Duan et al. [67] fabricated an impedimetric aptasensor employing Ti$_3$C$_2$T$_x$ MXene nanosheets patterned using iron phthalocyanine quantum dots (QDs) by $\pi - \pi^*$ stacking interactions with MXene for the determination of microRNA-155 (miRNA-155). After modification of the bare gold electrode (AuE, $R_{ct} \approx 109$ $\Omega$) with QDs, MXene nanosheets, and MXene@QD nanohybrid, the $R_{ct}$ values increased. $R_{ct}$ values of 1.09 k$\Omega$, 251 $\Omega$, and 251 $\Omega$ were obtained for the QD/AuE, MXene/AuE, and MXene@QDs/AuE electrodes, respectively. The $R_{ct}$ values of all electrodes increased after the attachment of complementary DNA (cDNA) strands and resulted in values of 2.18 k$\Omega$, 713 $\Omega$, and 483 $\Omega$ for the cDNA/QDs/AuE, cDNA/MXene/AuE, and cDNA/MXene@QD/AuE, respectively. This increase in $R_{ct}$ was caused by the presence of a negative charge of the cDNA strands on the

interface, inhibiting diffusion of a negatively charged $[Fe(CN)_6]^{3-/4-}$ probe toward the interface. A further increase in the $R_{ct}$ values confirmed the hybridization of cDNA with miRNA-155 with values of 2.28 kΩ, 853 Ω, and 1.11 kΩ for the miRNA155/cDNA/QDs/AuE, miRNA-155/cDNA/MXene/AuE, and miRNA155/cDNA/MXene@QD/AuE, respectively (Figure 12). The MXene@QD nanohybrid offered more anchoring sites for the immobilization of DNA due to the large effective surface area of the assembled nanostructure. The biosensor was used for detection of the analyte over a wide concentration range of 0.01–10,000 fM with LOD of 4.3 aM using EIS (Figure 13). The biosensor again exhibited good selectivity, assay reproducibility, and storage stability (Figure 13c,e) [67].

EIS proved to be an ultrasensitive platform of detection for biosensors based on immobilized DNA for detection of the protein using an aptamer-based biosensor with LOD of 15 aM (Section 4.3) [66] and for detection of miRNA using a DNA-based biosensor with LOD of 4.3 aM (Section 4.4) [67].

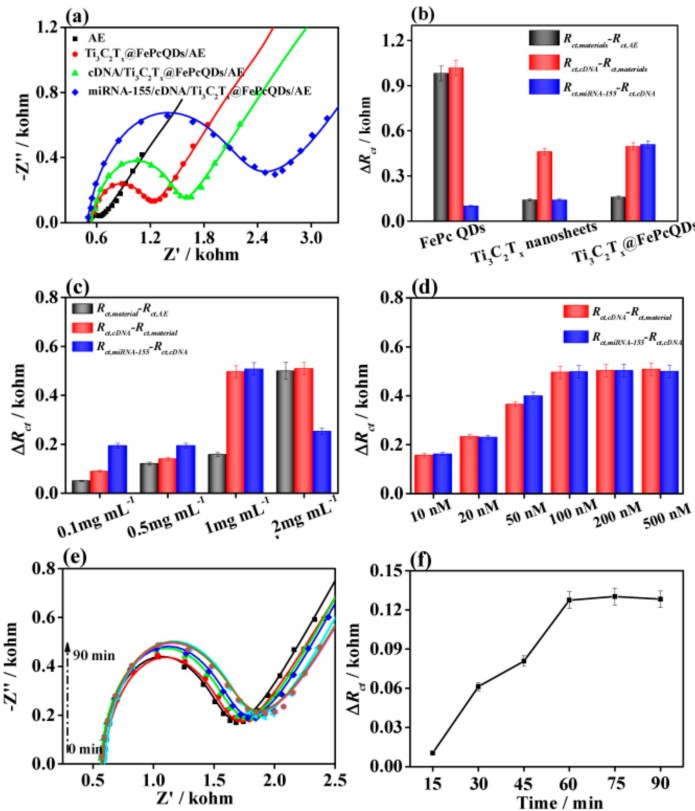

**Figure 12.** (**a**) Nyquist plots for various stages of development of the aptasensor for analysis of micro RNA-155 (miRNA-155) based on MXene@QD nanohybrid (QDs stands for quantum dots). (**b**) Variations of the $R_{ct}$ values for three different biosensor devices. (**c**) Variations of the $R_{ct}$ values for the biosensor device based on the MXene@QD nanohybrid with analyte concentrations from 0.1 to 2 mg mL$^{-1}$. (**d**) The $\Delta R_{ct}$ values for different aptamer concentrations from 10 to 500 nM for detection of 10 fM miRNA-155using the MXene@QD-based aptasensor. (**e**) The influence of binding time on detecting miRNA-155 using the MXene@QD-based aptasensor. Nyquist plots of the MXene@QD-based aptasensor incubated with miRNA-155 solution (10 fM) and (**f**) their corresponding $\Delta R_{ct}$ values. Reprinted from ref. [67], Copyright (2020), with permission from Elsevier.

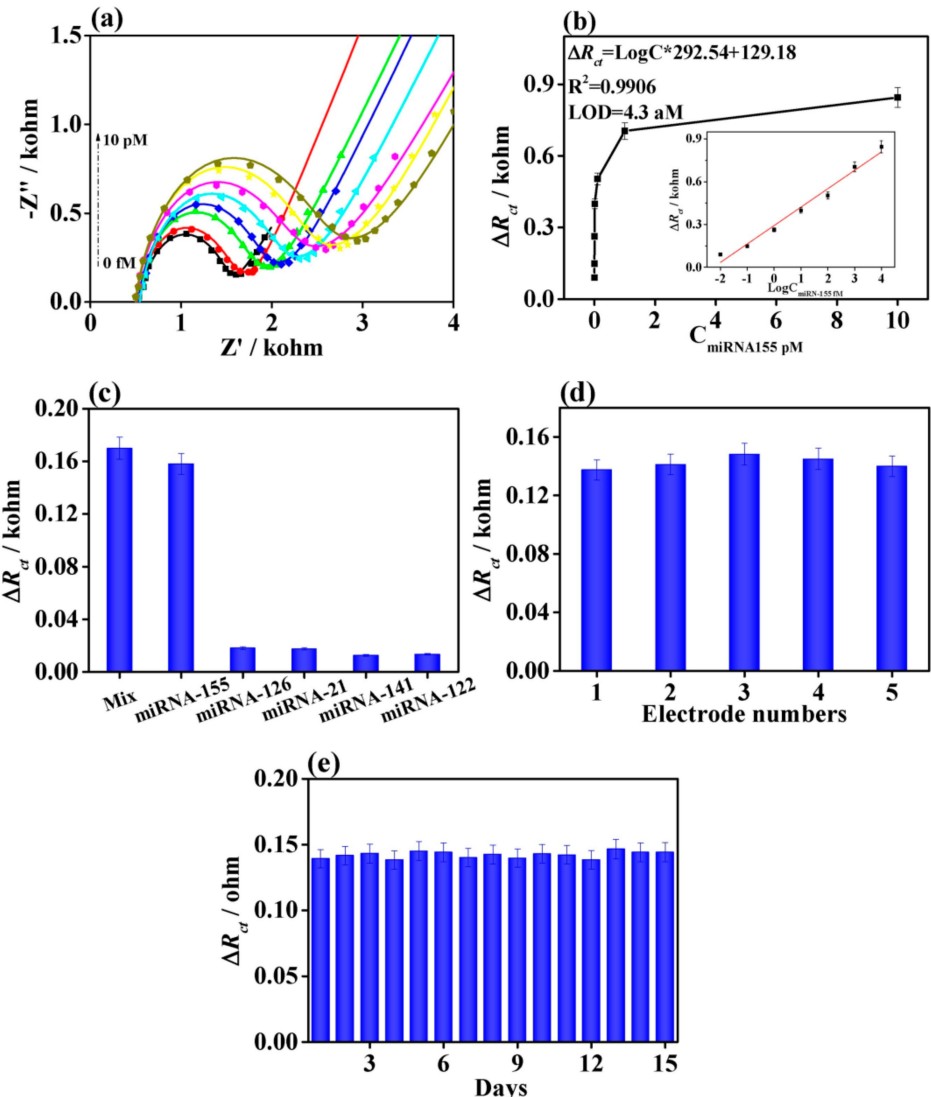

**Figure 13.** (**a**) Nyquist plots for detection of miRNA-155 from 0.01 to 10,000 fM using the MXene@QD-based aptasensor. (**b**) Dependence of $\Delta R_{ct}$ on the concentration of miRNA-155. (**c**) The $\Delta R_{ct}$ values of MXene@QD nanohybrid-based electrochemical aptasensor by separately adding the interferences with the concentration of 10 fM, miRNA-155 with 0.1 fM, and their mixture. (**d**) Reproducibility and (**e**) stability of the MXene@QD-based aptasensor for detecting miRNA-155 with the concentration of 0.1 fM. Reprinted from ref. [67], Copyright (2020), with permission from Elsevier.

### 4.5. EIS for Electrochemical Investigation of MXene-Modified Electrodes with Molecularly Imprinted Polymers

The amyloid-$\beta$ protein contains 39–43 amino acids with an abundance of two amyloid forms such as A$\beta_{40}$ or A$\beta_{42}$. Özcan with co-workers [68] investigated detection of the A$\beta_{42}$ protein to correlate it with a neurodegenerative disorder like Alzheimer disease *via* a molecularly imprinted polymer (MIP)-based biosensor based on d-Ti$_3$C$_2$T$_X$ MXene and MWCNTs composite. In an attempt to develop sensitive and selective sensors, a molecular imprinting technique was employed specifically to form cavities in the cross-linked polymeric matrix with specific recognition sites for the analyte. The analyte could be detected with LOD of 0.3 fg mL$^{-1}$ in a linear range of 1.0 fg mL$^{-1}$–100.0 fg mL$^{-1}$ using the device. The colloidal solutions of MXene and MWCNTs were mixed together in a mass ratio of 3:1 to obtain the MXene/MWCNTs' composite. The mixture consisting of pyrrole as a monomer and A$\beta_{42}$ as a template was used for the preparation of all MI*P*-modified electrodes (MIP/GCE, MIP/MXene/GCE, and MIP/MXene/MWCNTs/GCE) by applying the CV method with 25 cycles and a scan rate of 100 mV s$^{-1}$ at ambient temperature. The electrostatic interactions between the pyrrole

units and $A\beta_{42}$ in the matrix were broken using a desorption solution (1.0 M NaCl) in order to leach out the imprinted analyte to form the recognition cavities. Along with other analytical methods, EIS was also included to characterize MXene, MWCNTs, and MXene/MWCNTs-modified interfaces. The $R_{ct}$ values were of 135 $\Omega$, 90 $\Omega$, 52 $\Omega$, and 25 $\Omega$ for the bare GCE, MXene/GCE, MWCNTs/GCE, and MXene/MWCNTs/GCE, respectively. The value of $R_{ct}$ increased to 160 $\Omega$ after the preparation of MIP/MXene/MWCNTs/GCE with $A\beta_{42}$ protein present in the matrix. The $R_{ct}$ value decreased when the $A\beta_{42}$ protein was eluted. A good correlation between CV and EIS was observed when both methods were applied to the step-by-step characterization of the MIP-based sensor build up [68].

Kadirsoy et al. [69] also successfully used delaminated sulphur-doped MXene (S-Ti$_3$C$_2$T$_x$) in combination with a molecularly imprinted quartz crystal microbalance (QCM) sensor for the detection of chlorpyrifos, one of the most frequently used organophosphate insecticides. The MIP/MXene/QCM sensor was characterized by different microscopic and spectroscopic techniques. By applying this environmentally friendly sensor for chlorpyrifos detection, LOD of $3.0 \times 10^{-13}$ M was estimated (linear range $1.0 \times 10^{-12}$ M–$1.0 \times 10^{-13}$ M). The added value of MXene is in its 2D structure, which prevents restacking, thereby limiting access of the target molecules to the QCM chip surface. The charge transfer resistance ($R_{ct}$) values thus obtained were 175 $\Omega$, 75 $\Omega$, and 25 $\Omega$, for bare GCE, Ti$_3$C$_2$T$_x$ MXene/GCE, and d-S-Ti$_3$C$_2$T$_x$ MXene/GCE, respectively [69].

Finally, EIS was chosen for the characterization of several steps needed to develop MIP-based sensor devices used in detection of the high molecular mass analyte ($A\beta_{42}$ protein) [68] and low molecular mass analyte (insecticide) [69].

## 5. Conclusions

This review presents EIS as the tool of choice to characterize MXene-based interfaces directly in situ. EIS helped to elucidate the potential window under which MXene-patterned electrodes were stable and a mechanism behind MXene instability in the cathodic potential window. At the same time, EIS helped to characterize MXene interfaces using MXenes prepared *via* two distinct routes and helped understanding of the kinetics of spontaneous grafting of diazonium containing betaines onto MXene-patterned interfaces. Finally, EIS proved to be a very effective tool for monitoring the step-by-step building of biosensing interfaces for the successful construction of antibody- and aptamer-based biosensors and biosensor-integrated MIP layers. Although a good correlation between CV and EIS was observed when both methods were used to characterize the step-by-step build-up of various types of MXene-based biosensor, we would recommend using the EIS method. This is because, for EIS assays, the MXene interface is exposed to potentials, which are very close to open circuit potential or potentials, which do not induce oxidation of MXene. By contrast, for a successful run of CV, potentials significantly exceeding +200 mV *vs*. Ag/AgCl are needed with oxidation of the MXene negatively affecting such assays. Electrochemical methods, especially EIS, are fast, reliable, and sensitive to interfacial changes on a nanoscale level. In particular, electrochemistry can provide information on the redox stability of nanomaterials, stability against oxidation, and resistance to biofouling and it is possible to monitor the success of surface modification with nanomaterial. At the same time, it is worth noting that such changes can be monitored in situ and in real time if required.

**Author Contributions:** Conceptualisation, L.L. and J.T.; Formal analysis, M.B. and P.F.; Writing—original draft preparation, J.A., L.L., and J.T.; Writing—review and editing, J.A., L.L., M.B., P.F., and J.T.; Project administration, J.T.; Funding acquisition, J.T. All authors have read and agreed to the published version of the manuscript.

**Funding:** The authors wish to acknowledge the financial support received from the Innovative Training Network (No. 813120).

**Conflicts of Interest:** The authors declare no conflict of interest.

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
