# Peer review of "Electrochemical Impedance Spectroscopy on 2D Nanomaterial MXene Modified Interfaces: Application as a Characterization and Transducing Tool"

_chemosensors, doi:10.3390/chemosensors8040127_

Round 1
Reviewer 1 Report
The review article of Aguedo et al. covers the relevant and timely topic of electrochemical impedance spectroscopic analysis of novel MXene nanomaterials.
The literature review is fairly comprehensive and could be of interest to the field. However the manuscript requires significant editing to revise the language and style and address some major inconsistencies.
Below the specific comments:
- Many acronyms and symbols are introduced without being previously defined. Some examples: GCE, R(Q[RW]), LOD, MWCNTs, CV
-
References 15 and 16 are not the only relevant works on the topic of MXene toxicity. A number of in vitro and in vivo studies have been dedicated to the biocompatibility of different types of MXenes either in the form of loose 2D flakes or films. See for example all the literature on MXene for cancer theranostics, which includes extensive safety data, or recent work on MXene-based neural interfaces. In all of these studies several species of MXenes like Ti3C2, Nb2C, and Mo2C where found to be non-toxic, even to material and substrate sensitive cells, like neurons. The authors should expand the list of cited works. They should also carefully specify the MXene species for which toxicity data exist
- The list of applications of MXenes should include wearable and implantable sensors (like pressure and strain sensors, as well as neural microarrays)
- For the works where CV analysis has been conducted in conjunction with EIS, the authors should elaborate on how to interpret the CV data in relation to EIS
- The conclusion should include an outlook to future studies where electrochemistry could be crucial and potentially enabling of new applications/new investigations on MXenes
-
Figure 1: panel labeling in the figure and caption should proceed in progressive order
- There are several typos and incomplete sentences. Here some examples:
- page 1: broking
- page 2: Due to the fast evolution of the material, there is still a demand for improving their synthesis.
- page 2 and other instances: use of the singular form MXene, but then the verb is plural
- page 3: exposed [missing: to] electrochemical
- page 3: Ohm-ic
- page 3: equivalent [missing: circuit]
-
"The frequency of the AC signal applied to the system is varied with overall impedance of the cell recorded as a function of an input frequency." This sentence is not clear. Do the authors mean: "while impedance is monitored"?
- page 4: timescale. it should be "timescales"
- Zim =R. incorrect. It should be Zreal=R
- jomegaC missing in the definition of Zim
- page 4: describe. it should be "describes"
- "If the visual examination of the features of the Nyquist plot is 133 combined with the knowledge about the device allows creating an equivalent circuit model 134 describing a specific process by each circuit element represents". This sentence is unclear
Author Response
Reviewer 1
The review article of Aguedo et al. covers the relevant and timely topic of electrochemical impedance spectroscopic analysis of novel MXene nanomaterials.
The literature review is fairly comprehensive and could be of interest to the field. However the manuscript requires significant editing to revise the language and style and address some major inconsistencies.
Below the specific comments:
Many acronyms and symbols are introduced without being previously defined. Some examples: GCE, R(Q[RW]), LOD, MWCNTs, CV
- The acronyms and symbols are defined in the text, as required.
References 15 and 16 are not the only relevant works on the topic of MXene toxicity. A number of in vitro and in vivo studies have been dedicated to the biocompatibility of different types of MXenes either in the form of loose 2D flakes or films. See for example all the literature on MXene for cancer theranostics, which includes extensive safety data, or recent work on MXene-based neural interfaces. In all of these studies several species of MXenes like Ti3C2, Nb2C, and Mo2C where found to be non-toxic, even to material and substrate sensitive cells, like neurons. The authors should expand the list of cited works. They should also carefully specify the MXene species for which toxicity data exist.
- The paragraph mentioning prospect of MXenes applicable as theranostic agents is now included:
“The physiological stability and biocompatibility of 2D Nb2C nanosheets was achieved in combination with polyvinylpyrrolidone without any noticeable toxicity exhibited both in vitro and in vivo as well. It was also pointed out to the fact, that 2D MXenes are candidates of choice as theranostic agents for cancer diagnostics and therapeutics [18]. Other type of MXene i.e. Ti3C2 was also confirmed for in vitro/in vivo photothermal ablation of tumor using near-infrared irradiation [19]. In addition the authors demonstrated biocompatible soybean phospholipid (SP) modified Ta4C3 nanosheets [20] without any apparent toxicity, that were effectively employed for in vitro/in vivo photothermal ablation of tumor. Ti3C2Tx MXene was proven as a biocompatible and safe nanomaterial during examination of its toxicity using the zebrafish embryos as an in vivo model [21]. Multifunctional composite consisting of Ti3C2 patterned with paramagnetic MnOx [22] species can play an important role in theranostic medicine, e.g. for efficient magnetic resonance and photoacoustic dual-modality imaging-guided photothermal therapy to cure cancer. Nb2C quantum dots (QDs) [23] behave in a more biocompatible way to human umbilical vein endothelial cells (HUVECs) compared to Ti3C2 QDs showing cytotoxic reaction. 2D Ti2NTx nanosheets [24] exhibited higher toxicity towards cancerous (MCF-7 and A365) cell lines compared to normal ones with no toxic behavior to non-malignant (MCF-10A and HaCaT) cell lines.”
The list of applications of MXenes should include wearable and implantable sensors (like pressure and strain sensors, as well as neural microarrays).
- These types of sensors are now also mentioned amongst other applications of MXene:
“To date, 20+ different MXene compositions have been synthetized comprising of Ti3C2Tx, Ti2CTx, Ti2NT, Mo2CTx, and/or Nb4C3Tx [25], 70 varieties of MAX phases have been investigated and more MAX phases are predicted [2]. In the last ten years, MXenes have become a very attractive material for use in a variety of applications, including biosensors [26, 27], sensors with improved electrocatalytic activity [4, 28-30], wearable sensors [31-34], MXene-based flexible microelectrode arrays for recording of neural activity [35, 36], biomedical applications [37, 38], energy storage technologies [39], supercapacitors [40], batteries [41], hydrogen storage devices [42], catalytic devices [43], photocatalysts [44], field-effect transistors [45], water purification systems [46], devices for environmental remediation [47] and moreover due to their outstanding conductivity [48] they have been employed as an electromagnetic shielding material [49].“
For the works where CV analysis has been conducted in conjunction with EIS, the authors should elaborate on how to interpret the CV data in relation to EIS.
- The following text is now part of the Conclusion Section:
“Although, a good correlation between CV and EIS was seen, when both methods were applied for characterization of step by step build-up of various types of MXene-based biosensor, we would like to suggest using EIS method. The main reason behind that is the fact, that for EIS assays the MXene interface is exposed to potentials which are very close to open circuit potential or potentials, which due to not induce oxidation of MXene. Contrary, for successful run of CV, potentials significantly exceeding +200 mV vs. Ag/AgCl are needed with oxidation of the MXene, negatively affecting such assays.”
The conclusion should include an outlook to future studies where electrochemistry could be crucial and potentially enabling of new applications/new investigations on MXenes.
- The following text is part of the Conclusion section:
“Electrochemical methods and especially EIS are quick, reliable and sensitive to interfacial changes at a nanoscale level. In particular, electrochemistry can provide information about redox stability of nanomaterials, stability against oxidation, resistance towards biofouling and it is possible to monitor success of surface modification by nanomaterial. At the same time it is important to highlight that such changes can be monitored in-situ and in real time if required.”
Figure 1: panel labeling in the figure and caption should proceed in progressive order.
- This is done as requested.
There are several typos and incomplete sentences. Here some examples:
page 1: broking
- It was corrected, as it was required.
page 2: Due to the fast evolution of the material, there is still a demand for improving their synthesis.
- It was corrected, as it was required.
page 2 and other instances: use of the singular form MXene, but then the verb is plural
- It was corrected, as it was required.
page 3: exposed [missing: to] electrochemical
- It was corrected, as it was required.
page 3: Ohm-ic
- It was corrected, as it was required.
page 3: equivalent [missing: circuit]
- It was corrected, as it was required.
"The frequency of the AC signal applied to the system is varied with overall impedance of the cell recorded as a function of an input frequency." This sentence is not clear. Do the authors mean: "while impedance is monitored"?
- This sentence war rewritten:
“The frequency of the AC signal applied to the system is varied with overall impedance of the cell recorded as a function of an input frequency, while impedance is monitored. In this way it is possible to identify processes occurring on different timescales and to separate electronic from ionic processes in mixed conductors.”
page 4: timescale. it should be "timescales"
Zim =R. incorrect. It should be Zreal=R
jomegaC missing in the definition of Zim
page 4: describe. it should be "describes"
- It was corrected, as it was required.
"If the visual examination of the features of the Nyquist plot is combined with the knowledge about the device allows creating an equivalent circuit model describing a specific process by each circuit element represents". This sentence is unclear
- This sentence was rewritten:
“Visual examination of the Nyquist plot combined with the knowledge about the device allows for an equivalent circuit model to be set up accurately, correctly describing all processes involved (Figs. 3e, 3f, 3g). Data present in the Nyquist plot close to the origin are associated with high frequency spectrum.”

Reviewer 2 Report
This review “Electrochemical Impedance Spectroscopy on MXene modified interfaces. Application as a characterization and a transducing tool” is compact and concise. The authors need to address a few concerns.
- 1st part should be little more elaborative and thus some recent and interesting references should be added i.e.
2D Transition Metal Carbides (MXene) for Electrochemical Sensing: A Review, Critical Reviews in Analytical Chemistry, 2020
Electromagnetic interference shielding with 2D transition metal carbides (MXenes), 2016, Science 353 (6304), 1137-1140
2D Transition Metal Carbides (MXenes): Applications as an Electrically Conducting Material, 2020, Advanced Materials, 2002159
Anomalous absorption of electromagnetic waves by 2D transition metal carbonitride Ti3CNTx (MXene), 2020, Science 369 (6502), 446-450.
- Current challenges and future perspectives should be highlighted
- A critical analysis of each subsection in section 2 would enhance the quality of this work.
Author Response
Reviewer 2
Comments and Suggestions for Authors
This review “Electrochemical Impedance Spectroscopy on MXene modified interfaces. Application as a characterization and a transducing tool” is compact and concise. The authors need to address a few concerns.
- 1st part should be little more elaborative and thus some recent and interesting references should be added i.e.
2D Transition Metal Carbides (MXene) for Electrochemical Sensing: A Review, Critical Reviews in Analytical Chemistry, 2020
Electromagnetic interference shielding with 2D transition metal carbides (MXenes), 2016, Science 353 (6304), 1137-1140
2D Transition Metal Carbides (MXenes): Applications as an Electrically Conducting Material, 2020, Advanced Materials, 2002159
Anomalous absorption of electromagnetic waves by 2D transition metal carbonitride Ti3CNTx (MXene), 2020, Science 369 (6502), 446-450.
- The papers were cited in the introduction part.
- Current challenges and future perspectives should be highlighted
- The following sentence was added into Section 2:
“This is possible due to the fact that electrochemical methods are quick, reliable and sensitive to interfacial changes at a nanoscale level. In particular, electrochemistry can provide information about redox stability of nanomaterials, stability against oxidation, resistance towards biofouling and it is possible to monitor success of surface modification by nanomaterial. At the same time it is important to highlight that such changes can be monitored in-situ and in real time if required.”
- A critical analysis of each subsection in section 2 would enhance the quality of this work.
- Each section is accompanied with a short paragraph in a form of a conclusion.
Section 3.1:
“We can conclude that EIS is an effective tool for in-situ characterization of charge transfer resistance (or conductivity) of MXene modified interfaces applied as sensors. In particular, EIS can provide information if MXene was exposed to potential leading to MXene oxidation, but it can also information if it is worth to combine MXene with other types of nanomaterials in order to make such hybrid nanoparticle modified interface more conductive.”
Section 3.2:
“EIS investigation was successfully applied to better understand interfacial properties of MXenes prepared in two distinct routes. In particular EIS provided insights about initial redox conditions, but also about density of negative charge on MXenes prepared using two routes.”
Section 3.3:
“EIS investigation indicated an effective pairing of the anionic and cationic groups within SB layer resulting in low resistance towards a ferri/ferro redox couple. The study also showed, that zwitterion-modified MXene prepared by electrochemical grafting demonstrated higher Rct values in comparison to spontaneous process for CB (1,610 Ω vs. 838 Ω) and for SB (228 Ω vs. 88 Ω), as well. It was proved, that the higher density of zwitterions on the interface was induced electrochemically in comparison to the process based on spontaneous grafting [13].”
Section 4.1:
“Two electrochemical methods (EIS and CV) were applied to follow build-up of the biosensing interface [62]. Both methods provided the same conclusion that modification of GCE by CS-MXene made the system more conductive, while after immobilization of the enzyme the conductivity of the interface slightly dropped (Fig. 6). The latter really confirmed successful immobilization of the enzyme on the interface.”
Section 4.2:
“EIS (Fig. 8B) as tool of choice to examine interfacial properties of modified conductive layers was also independently confirmed by CV (Fig. 8A and 8C) [64]. As can be clearly seen from the Nyquist diagram, Rct decreased after modification of GCE with delaminated sulfur doped MXene (i.e. d-S-MXene/GCE) and AuNPs/d-S-MXene/GCE in comparison to bare GCE. The Rct increased after immobilization of Ab1 on AuNPs/d-S-Ti3C2Tx MXene/GCE. After blocking by BSA, Rct increased and the Rct further increased after incubation with the analyte (PCT) [64].”
Section 4.3 and Section 4.4:
“EIS proved to be an ultrasensitive platform of detection for the biosensors based on immobilized DNA for detection of the protein using aptamer-based biosensor with LOD of 15 aM (Section 4.3) [65] and for detection of miRNA using DNA-based biosensor with LOD of 4.3 aM (Section 4.4) [66].”
Section 4.5:
“Finally, the EIS was chosen for characterization of several steps needed for development of MIP-based sensor devices applied for detection of high molecular weight analyte (Aβ42 protein) [67] and low molecular weight analyte (insecticide) [68].”
Conclusion section:
“Although, a good correlation between CV and EIS was seen, when both methods were applied for characterization of step by step build-up of various types of MXene-based biosensor, we would like to suggest using EIS method. The main reason behind that is the fact, that for EIS assays the MXene interface is exposed to potentials which are very close to open circuit potential or potentials, which due to not induce oxidation of MXene. Contrary, for successful run of CV, potentials significantly exceeding +200 mV vs. Ag/AgCl are needed with oxidation of the MXene, negatively affecting such assays. Electrochemical methods and especially EIS are quick, reliable and sensitive to interfacial changes at a nanoscale level. In particular, electrochemistry can provide information about redox stability of nanomaterials, stability against oxidation, resistance towards biofouling and it is possible to monitor success of surface modification by nanomaterial. At the same time it is important to highlight that such changes can be monitored in-situ and in real time if required.”

Round 2
Reviewer 1 Report
The authors have adequately addressed all the reviewer comments.
Extensive revision the English language and style is still required. Proofreading by a native speaker is advisable.
Author Response
Dear Reviewer,
extensive edition of English was done by a native English speaker.